# Synaptic Vesicle Protein 2A Expression in Glutamatergic Terminals Is Associated with the Response to Levetiracetam Treatment

**DOI:** 10.3390/brainsci11050531

**Published:** 2021-04-23

**Authors:** Itzel Jatziri Contreras-García, Gisela Gómez-Lira, Bryan Víctor Phillips-Farfán, Luz Adriana Pichardo-Macías, Mercedes Edna García-Cruz, Juan Luis Chávez-Pacheco, Julieta G. Mendoza-Torreblanca

**Affiliations:** 1Área de Neurociencias, Biología de la Reproducción, Unidad Iztapalapa, Universidad Autónoma Metropolitana, Ciudad de México 09340, Mexico; jatziri1984@hotmail.com; 2Departamento de Farmacobiología, Centro de Investigación y Estudios Avanzados del Instituto Politécnico Nacional, Ciudad de México 14330, Mexico; glira@cinvestav.mx; 3Laboratorio de Nutrición Experimental, Subdirección de Medicina Experimental, Instituto Nacional de Pediatría, Ciudad de México 04530, Mexico; bvphillips@yahoo.com; 4Departamento de Fisiología, Instituto Politécnico Nacional, Escuela Nacional de Ciencias Biológicas, Ciudad de México 07738, Mexico; lpichardom@ipn.mx; 5Laboratorio de Neurociencias, Subdirección de Medicina Experimental, Instituto Nacional de Pediatría, Ciudad de México 04530, Mexico; ednagcmeg@gmail.com; 6Laboratorio de Farmacología, Subdirección de Medicina Experimental, Instituto Nacional de Pediatría, Ciudad de México 04530, Mexico; jchavez_pacheco@hotmail.com

**Keywords:** SV2A, levetiracetam, pharmacoresistance, VGAT, VGLUT, temporal lobe epilepsy, hippocampus

## Abstract

Synaptic vesicle protein 2A (SV2A), the target of the antiepileptic drug levetiracetam (LEV), is expressed ubiquitously in all synaptic terminals. Its levels decrease in patients and animal models of epilepsy. Thus, changes in SV2A expression could be a critical factor in the response to LEV. Epilepsy is characterized by an imbalance between excitation and inhibition, hence SV2A levels in particular terminals could also influence the LEV response. SV2A expression was analyzed in the epileptic hippocampus of rats which responded or not to LEV, to clarify if changes in SV2A alone or together with glutamatergic or GABAergic markers may predict LEV resistance. Wistar rats were administered saline (control) or pilocarpine to induce epilepsy. These groups were subdivided into untreated or LEV-treated groups. All epileptic rats were video-monitored to assess their number of seizures. Epileptic rats with an important seizure reduction (>50%) were classified as responders. SV2A, vesicular γ-aminobutyric acid transporter and vesicular glutamate transporter (VGLUT) expression were assessed by immunostaining. SV2A expression was not modified during epilepsy. However, responders showed ≈55% SV2A-VGLUT co-expression in comparison with the non-responder group (≈40%). Thus, SV2A expression in glutamatergic terminals may be important for the response to LEV treatment.

## 1. Introduction

Synaptic vesicle protein 2A (SV2A) is an essential membrane protein, universal to all types of neuronal terminals [1,2]. Its exact physiological function is not completely known. However, some reports show that SV2A participates in the synaptic vesicle cycle. It may be involved in vesicular priming [2,3], enhancing the probability of neurotransmitter release at presynaptic terminals [4] and regulating action potential calcium-dependent neurotransmitter release [5,6]. SV2A also participates in the modulation of endocytosis by regulating the expression and trafficking of synaptotagmin (a Ca^2+^ sensor protein) after synaptic vesicle fusion [7]. In this line, SV2A stimulates AP2 clathrin adaptor binding to synaptotagmin; consolidating the endocytosis complex, thereby allowing synaptotagmin internalization and trafficking. This, in turn, maintains the readily releasable pool [2,6,7,8]. Later, SV2A-synaptotagmin binding prepares vesicles for calcium-induced fusion [2,6]. Finally, SV2A is intimately related to epilepsy, as indicated by several findings: (i) animals lacking SV2A fail to grow, exhibit severe seizures and die within 3 weeks (ii) heterozygous mice show a pro-epileptic profile in the 6 Hz, pentylenetetrazol, kainate, pilocarpine and kindling models [9,10]; (iii) missense mutations (L^174Q^ and R^383Q^) of the *Sv2a* gene increase seizure susceptibility in rats and cause intractable epilepsy in patients, respectively [11,12]; (iv) SV2A is the target of the antiepileptic drugs levetiracetam (LEV) and brivaracetam [1,13,14], constituting their primary mechanism of action.

Under physiological conditions, SV2A has a differential expression pattern throughout the brain [15,16,17]. This pattern is altered during epilepsy, as shown by clinical and preclinical studies. In patients with intractable temporal lobe epilepsy (TLE) and hippocampal sclerosis, SV2A levels are reduced in the hippocampus [18] and anterior temporal neocortex [19]. Apparently, this decrease occurs in areas with neuronal loss [20].Likewise, SV2A expression is down-regulated in patients with intractable epilepsy due to focal cortical dysplasia and cortical tubers [21]. In some animal models of epilepsy, SV2A levels are decreased in the hippocampus; which is particularly epileptogenic, making it the main altered anatomical substrate of TLE. SV2A expression is reduced in the hippocampus throughout the latent period and during chronic epilepsy [18]. Similar results are seen in a genetic model of spontaneously epileptic rats [22]. In contrast, increased SV2A levels are observed in some hippocampal layers using a kindling model [23,24] and due to status epilepticus (SE) in the pilocarpine model of TLE [25].

Changes in SV2A protein expression induced during epilepsy could be a critical factor in the response to LEV treatment, since SV2A is its molecular target. Indeed, in patients with glioma and seizures, the effectiveness of the clinical response to LEV has been associated with SV2A expression levels [26]. Additionally, a microarray analysis of the hippocampus in epileptic patients not responding to LEV treatment shows overexpression of *Sv2a* and other genes encoding proteins involved in vesicle trafficking. This suggests that adequate expression levels of these proteins must be maintained for proper endocytosis, as well as LEV action [27]. Taken together, these results indicate that any change in SV2A expression may participate in the pathogenesis and treatment of epilepsy. Furthermore, changes in SV2A expression might explain the resistance to LEV treatment in some epileptic patients. Thus, an extensive analysis of SV2A expression by immunostaining was performed in the hippocampus of rats that responded or not to LEV treatment, using the pilocarpine model of TLE, to clarify if SV2A changes may predict pharmacoresistance to LEV.

On the other hand, although SV2A is found in both GABAergic and glutamatergic terminals, several studies show that it functions preferentially within inhibitory synapses [17,28,29]. In different stages of epilepsy, SV2A is co-expressed generally with GABAergic markers, suggesting a preferential expression of SV2A by GABAergic interneurons [12,25,30,31,32]. Hippocampal slice recordings from SV2A knockouts show a more pronounced effect on inhibitory postsynaptic currents [9,33]. Similarly, the Sv2a^L174Q^ mutation preferentially reduces vesicular GABA release, not glutamate release, in the hippocampus and amygdala [12,28,32]. Moreover, LEV treatment also preferentially augments vesicular GABA release [34]. Because epilepsy is characterized by an imbalance of excitatory to inhibitory systems and since SV2A is the main molecular target of LEV, whether GABAergic or glutamatergic terminals co-express SV2A could be decisive for LEV resistance. Thus, we also studied the association of SV2A with inhibitory or excitatory terminals to assess whether the response to LEV treatment is associated with a specific type of terminal.

## 2. Methods

### 2.1. Animals

Three-month-old male Wistar rats (250–300 g) were purchased from Envigo (Indianapolis, IN, USA) and housed using a controlled temperature (22 ± 2 °C) and a 12 h dark-light cycle (lights on at 6:00 a.m.), with food and water ad libitum. All procedures were in line with the NIH Guide for the Care and Use of Experimental Animals and the Mexican law (SAGARPA NOM-062-ZOO-1999). All protocols were approved by our Institutional Animal Welfare Committee (INP-064-2015).

### 2.2. Groups

The animals were randomly distributed into 4 groups: (1) a control group, (2) a control group treated with LEV via an implanted osmotic pump, (3) an epileptic group and (4) an epileptic group treated with LEV. Previously, a marked inter-individual variability to LEV was observed in the pilocarpine model of TLE, generating responder and non-responder rats. This variation is not associated with pharmacokinetic differences [35]. Thus, we evaluated seizure number before and during treatment to subdivide group 4 according to their response. If the animals showed an important seizure reduction (>50%), they were classified as responders; otherwise (<50% seizure reduction), they were classified as non-responders. Table 1 shows the conditions for each group and Figure 1 describes the experimental design.

### 2.3. Induction of SE and Post-SE Care

On day 0 (Figure 1), the epileptic groups (3 and 4) were administered lithium chloride (127 mg/kg i.p.; Sigma-Aldrich, #L9650; Darmstadt, Germany). On day 1 (19 h after), rats were injected with methyl scopolamine bromide (1 mg/kg i.p.; Sigma-Aldrich, #S8502), and 30 min later status epilepticus (SE) was induced with pilocarpine hydrochloride (30 mg/kg i.p.; Sigma-Aldrich, #P6503). Their convulsive behavior was evaluated with the Racine scale [36]. SE was defined as continuous tonic-clonic seizures lasting more than 2 min with no recovery. 90 min after, animals were injected a single dose of diazepam (5 mg/kg i.m.; PISA; 070M95; Guadalajara, México) to stop SE. Immediately after, the rats were kept for one hour on an ice bed. A second dose of diazepam (5 mg/kg i.m.) was administered 8 h after the first injection. Additionally, rats were administered 0.9% NaCl (5 mL s.c.; PISA, 82175) to rehydrate them and housed at 17 °C overnight. All rats were then returned to controlled conditions and fed a nutritional supplement until they ate standard pellets on their own. The animals that did not present SE were eliminated from the study. The survival of rats that showed SE was 100%.

### 2.4. Behavioral Video-Monitoring of Seizures

To evaluate seizure number before and during LEV treatment, the EPI, R, and NR groups were video-monitored for 10 h during the light period (8:00–18:00; [37]) from day 42 until day 56 after SE (Figure 1). Rats were kept in individual acrylic cages. Spontaneous recurrent seizures (SRS) were scored if generalized convulsive seizures were observed (3–5 in the Racine scale). The videos were obtained with cameras (Steren, CCTV-970; CDMX, México) and viewed with the H.264 PlayBack program for Windows (v.1.0.1.15, Infinova, Guangdong, China. [34]).

### 2.5. Levetiracetam Treatment

Osmotic pumps (Alzet, 2ML1; release rate 10 μL/h; Durect Corporation; Cupertino, CA, USA) were implanted in treated groups (2 and 4) on day 49 (Figure 1). LEV was extracted from 2 tablets (1000 mg; Ultra Laboratorios, 226M2009; Guadalajara, México) and dissolved in 3 mL of 0.9% NaCl. The solution was sonicated for 10 min, centrifuged (Hettich, Mikro 12–24, type 2070-01, w/24; Westphalia, Germany) for 15 min at 1960 g and the supernatant was filtered (0.45 μm; Corning^TM^, #431220; Corning, NY, USA) before use. Prior to implantation, osmotic pumps were filled with LEV and incubated for 5 h at 37 °C in 0.9% NaCl. After this, rats were anesthetized with isoflurane (Sofloran^®^Vet, PISA, Q-7833-222), an incision was made at the level of the scapula and the pump was implanted subcutaneously. The pump continuously released LEV (≈300 mg/kg/day) for 1 week. Right after, rats were administered an acute dose of LEV (200 mg/kg i.p.; UCB Laboratories, 038M2010; Brussels, Belgium).

LEV blood concentration was measured on day 7 after implanting the osmotic pump (Figure 1). Sample blood (50 μL) was collected from the caudal vein and placed on a Guthrie card (Whatman^®^903; #10534612; Washington, D.C., USA [34]). LEV was extracted from the Guthrie card and analyzed by high-pressure liquid chromatography [38].

The number of seizures was recorded before and during treatment to evaluate the response to LEV. It was considered effective if seizures decreased more than 50% during treatment, any less was defined as no efficacy [26]. Only animals that presented 3 or more seizures before treatment were included.

### 2.6. Tissue Sample Collection and Processing

On day 56, all rats were deeply anesthetized and transcardially perfused with 0.9% NaCl followed by 4% paraformaldehyde (Sigma-Aldrich, #158127). The brains were removed and post-fixed in 4% paraformaldehyde for 24 h, subsequently cryoprotected in sucrose (Sigma-Aldrich, #S7903) and finally frozen with pre-chilled 2-methylbutane (Sigma-Aldrich, #270342). Serial sagittal hippocampal slices (40 μm thick) were obtained using a cryostat (Leica, Cryocut 1800; Wetzlar, Germany) and stored in a cryoprotectant solution until use [25].

### 2.7. Immunohistochemistry and Triple Immunofluorescence

Immunohistochemistry was performed on free-floating sections, all groups were processed simultaneously using the same conditions [25]. Briefly, sections were washed 3 times with phosphate buffer (PB 0.1 M, pH 7.4) made with monobasic and dibasic phosphate (Sigma-Aldrich, #S8282 and #S7907) for 10 min. Then, the tissue was incubated for 10 min in 3% hydrogen peroxide (Sigma-Aldrich, #H1009). After 3 washes with PB, slices were incubated in 5% fetal calf serum (Gibco, #16000-044; Waltham, MA, USA) for 1 h. Next, the sections were incubated overnight with SV2A antibody (1:500, mouse monoclonal E-8, Santa Cruz, #sc-376234; Dallas, TX, USA) diluted in fetal calf serum. The next day, the tissue was washed and incubated for 2 h with biotinylated horse anti-mouse IgG (1:500, Vector, #BA-2000; San Diego, CA, USA) diluted in fetal calf serum. Then, the sections were washed and incubated with avidin-biotinylated peroxidase (Vector, #PK-6100) for 45 min to amplify the signal. Finally, slices were washed, and the staining was revealed with 3,3′-diaminobenzidine (Vector, #SK-4100). The tissue was mounted onto gelatin-coated slides.

For triple immunofluorescence, slices contiguous to those used for immunohistochemistry were processed. This process was carried out simultaneously in all groups using the same conditions [17,29]. In brief, tissues were incubated overnight with the antibodies: SV2A (1:250, goat IgG; Santa Cruz, #sc-11936), vesicular γ-aminobutyric acid transporter (VGAT; inhibitory GABAergic terminals; 1:500, rabbit IgG, Sigma, #V5764) and vesicular glutamate transporter 1 (VGLUT; excitatory glutamatergic terminals; 1:1000, mouse IgG, Santa Cruz, #sc-377425) diluted in 5% bovine serum albumin (Sigma-Aldrich, #A9418;). The following day, the slices were washed and incubated with their respective secondary antibodies: donkey Alexa Fluor-647 anti-goat (1:200, Molecular Probes, #A-21447; Waltham, MA, USA), chicken Alexa Fluor-568 anti-rabbit (1:250, Molecular Probes, #A10042) and horse Dylight 488 anti-mouse (1:500, Vector, #410208) for 2 h. The tissues were mounted using Vectashield mounting medium (Vector, #H-1000).

### 2.8. SV2A Immunoreactive Measurement

Six randomly chosen sections representing the whole hippocampus were used to analyze SV2A immunohistochemistry. Images of these sections were obtained using an Olympus^®^ BX51 (Tokyo, Japan) microscope with an x4 objective connected to a digital video camera (MBF Bioscience, CX9000; Williston, ND, USA). SV2A optical density (OD) was evaluated, after calibration and background subtraction, using ImageJ v1.43 (Bethesda, MD, USA). We analyzed nine layers in the dorsal hippocampus: the molecular layer (Mol), granular layer (Gr), hilus, stratum radiatum (Rad) of CA3 and CA1, pyramidal layers (Pyr) of CA3 and CA1, as well as stratum oriens of CA3 and CA1.

For immunofluorescence analysis, 3 slices were chosen at random and 3 photographs were obtained for each layer. The images were captured with a x63/1.40 Oil DIC M27 objective of a confocal microscope (Carl Zeiss, LSM 800 with Airyscan; Stuttgart, Germany). They were analyzed with Fiji (1.52p, Bethesda, MD, USA) and the synapse counter plugin. SV2A, VGAT, and VGLUT positive puncta were counted, expressed as puncta number per area (50.7 × 50.7 μm). The spatial overlap (henceforth termed as co-localization or co-expression) among SV2A-VGAT or SV2A-VGLUT was detected and quantified (Figure 2).

### 2.9. Statistical Analysis

The number of SRS was expressed as medians and 25th–75th percentiles. The number of seizures in treated epileptic rats were compared with Mann–Whitney rank sum and Wilcoxon signed-rank tests. SV2A OD, immunofluorescence and LEV concentration in blood were expressed as mean ± SD, multiple comparisons were performed using one-way analysis of variance followed by Tukey tests. A p-value less than 0.05 was considered statistically significant. The analysis was done with GraphPad Prism for Windows (v. 9.0.0, San Diego, CA, USA).

## 3. Results

This study analyzed SV2A expression in the hippocampus of C, C+LEV, EPI, and EPI+LEV rats, in order to determine if alterations in SV2A protein expression associated to epilepsy may be important for LEV effectiveness. First, the results showed that epileptic rats have at least 3 seizures per week. In the EPI+LEV group, the number of SRS was dramatically reduced in some rats but not in others (as previously observed by [35]). Thus, we took advantage of this fact and divided this group into responders and non-responders (Table 2). Responder rats had an 89% decrease in seizures and non-responder rats only showed a 14.5% decrease.

Because the number of seizures could have changed due to differences in systemic LEV levels, its serum concentrations were quantified in osmotic-pump implanted rats. This was done at the end of the experiment (day 7 after implantation = day 56) in the C+LEV, R and NR groups. These groups showed no significant differences in LEV serum levels (Table 3).

SV2A OD was measured as a proxy for SV2A expression throughout the dorsal hippocampus (9 layers) in all groups. Surprisingly, no differences were observed in SV2A OD among the groups, including the two untreated groups: C and EPI (groups 1 and 3, data not shown) in any hippocampal layer examined (Figure 3). Minor changes in protein expression are observed in the R and NR groups. In the R group, SV2A OD increases slightly in the Mol and CA3 Rad layers compared to the C+LEV group (Figure 3A,F, respectively). On the other hand, a small increase in SV2A-immunoreactivity is also observed in the Mol layer (Figure 3A) of the NR group compared to the C+LEV group; while a minor decrease of SV2A OD is observed in the hilus, CA1 oriens, CA1 Pyr, and CA1 Rad layers (Figure 3G–I).

To analyze the possible role of SV2A in response to LEV treatment in GABAergic or glutamatergic terminals; SV2A, VGAT, and VGLUT expression was quantified. For this, the number of immunofluorescent puncta for VGAT, VGLUT, and their co-localization with SV2A was measured (as shown in Figure 2) in the same dorsal hippocampal layers as before.

Similarly to what was found before, no significant differences were observed in SV2A puncta number among the groups. However, differences in VGLUT and VGAT puncta were found in four hippocampal layers of rats that responded to LEV treatment compared with non-responders (Figure 4 and Figure 5). In the dentate gyrus Gr layer, VGLUT puncta decreased in NR rats compared to C+LEV and R animals. Similarly, SV2A-VGLUT co-localized puncta were reduced in the NR group compared to C+LEV rats (Figure 4A). VGLUT puncta increased significantly in R animals compared to C+LEV rats in the dentate gyrus hilus. In addition, SV2A-VGLUT co-localized puncta were higher in the R group than in NR rats (Figure 4B). In the Rad of CA3, VGLUT puncta increased significantly in R animals compared to C+LEV and NR rats. In addition, SV2A-VGLUT co-localized puncta were increased in the R group compared to NR animals (Figure 5A). Finally, in the CA1 oriens layer, VGLUT was higher in the R group compared to the C+LEV and NR groups. In contrast, reduced VGLUT puncta was seen in the NR group compared to C+LEV animals. Moreover, SV2A-VGLUT puncta was decreased in NR rats compared to the R and C+LEV groups (Figure 5B). In this layer, VGAT puncta decreased significantly in R and NR rats compared to C+LEV animals. This was reflected in SV2A-VGAT co-localized puncta (Figure 5B). Notice that VGAT and SV2A-VGAT puncta were not significantly different in the granular, hilus and CA3 Rad layers.

Another important observation was a significant association of SV2A with the inhibitory system, since 90–100% of GABAergic terminals in the principal cell layers (CA1 and CA3 Pyr, as well as the Gr) expressed SV2A (e.g., Figure 4A; VGAT vs. SV2A-VGAT). Synaptic layers (CA3 Rad, CA1 oriens, CA1 Rad, Mol and hilus) showed 75-85% of all GABAergic terminals co-expressing SV2A (e.g., Figure 5A,B; VGAT vs. SV2A-VGAT). By contrast, only 40–50% of all glutamatergic cells were associated with SV2A (e.g., Figure 4A and Figure 5A,B; VGLUT vs. SV2A-VGLUT); except in the Pyr CA3 and hilus, which had 65–80% co-localization.

## 4. Discussion

LEV is effective for seizure control in patients [39,40] and different animal models of epilepsy [41,42]. However, approximately 30% of patients do not respond to LEV treatment, despite having adequate therapeutic blood levels [27]. SV2A is an attractive target to explain the lack of response to LEV treatment, since LEV-SV2A binding constitutes the main mechanism for its antiepileptic effect [1,41]. However, there is little known about the expression of the SV2A protein and the response to treatment with levetiracetam. Some studies suggest that decreased SV2A expression could be a critical factor for the lack of response to LEV treatment [26,42]. A decrease in SV2A expression was observed in non-responder patients [26]. Herein, no significant changes in total SV2A expression were observed during the initial stage of epilepsy-induced by pilocarpine. Nonetheless, SV2A-VGLUT co-expression was decreased in the hilus, CA3 Rad and CA1 oriens of the non-responder group. Thus, SV2A levels in glutamatergic terminals might be important for the response to LEV treatment in an early phase of TLE.

TLE is characterized by progressive neuronal death in diverse brain regions, particularly the hippocampal formation [43,44]. This leads to synapse reorganization and marked proliferation of reactive astrocytes. Both these processes worsen in parallel with disease advancement [45,46]. Neuron loss and circuit remodelling alter the distribution and number of synaptic contacts as well as proteins, including SV2A [27,47]. Herein, SV2A quantification was performed in an initial chronic stage of epilepsy (7 weeks after SE). At this time, neuronal loss and reorganization are likely not drastic enough to affect SV2A expression. Hence, it would be interesting to study SV2A expression and cellular distribution in R and NR epileptic rats throughout all stages of the disease to determine if or when SV2A changes.

SV2A expression levels have an important influence on LEV efficacy. Its anticonvulsant activity, evaluated by the ability to increase seizure threshold in the 6 Hz seizure model, was significantly reduced (~50%) in SV2A +/− heterozygous mice. This reduction was related to 50% less LEV-binding sites in these mice [10]. Furthermore, LEV had no effect in hippocampal neurons lacking SV2A, consistent with the idea that the drug needs SV2A to act [48]. It is important to highlight that the reduction of SV2A in heterozygous or knock-out mice occurred in the entire brain, affecting all synaptic terminals in the same way. This drastic decrease does not occur in patients or in epileptic animals, since it has been reported that neuronal damage is selective [49,50]. On the other hand, several reports show that increased expression of synaptic components, including SV2A, results in synaptic dysfunction [23,24,27,30,48]. This disrupts LEV entry into vesicles [51] and thus decreases its effect [27]. Together, these findings indicate that there must be a basal level of SV2A expression for LEV to have an effect. Herein, no changes in SV2A expression were observed. However, modifications in SV2A distribution were seen in excitatory versus inhibitory terminals, mainly in the NR group. This could be associated with LEV pharmacoresistance in non-responder rats.

In this study, VGLUT increased in the hilus, CA3 Rad and CA1 oriens of R animals, but decreased in the Gr layer and CA1 oriens of NR rats. These changes may be partially a consequence of neuronal excitatory rearrangements resulting from neuronal death during epileptogenesis [45]. This reorganization occurs throughout the hippocampus; for example, granular neurons lose their synaptic contacts with hilar mossy cells, GABAergic interneurons and CA3 pyramidal cells. Their axons retract and generate aberrant connections with granule cell dendrites (of other cells or their own; mossy fiber sprouting), as well as with the remaining hilar interneurons [52]. In addition, differential neuronal death and synaptic rearrangements (including new dendrites and an increased number of extrinsic contacts from the amygdala and septum) have been seen in the CA3 Rad [47,53]. Finally, axon terminals in the oriens (Schaffer collaterals) increase if CA1 neurons are partly lost, in association with more axonal large-boutons [47]. Thus, these changes may partially explain the observed alterations in VGLUT expression.

SV2A-VGLUT co-expression was decreased in the Gr, hilus, CA3 Rad and CA1 oriens of NR animals, suggesting that SV2A expression in glutamatergic terminals is important for the response to LEV treatment. It has been proposed that LEV exerts its action by entering the intraluminal face of synaptic vesicles during recycling and endocytosis. This process may be slow due to the hydrophilic nature of LEV [51,54]. LEV-SV2A binding results in the production of fewer primed synaptic vesicles, a smaller readily-releasable pool and decreased post-synaptic potentials; all this suggests reduced neurotransmission [51]. As mentioned before, too little or too much SV2A expression decreases the effect of LEV. In this line, basal levels of SV2A-LEV binding are required to affect most neurons and diminish neurotransmission. If there is fewer SV2A in excitatory terminals of NR animals, there are fewer targets available for LEV action and thus a lesser effect of this drug. In contrast, in R animals, SV2A-VGLUT expression is maintained in excitatory terminals, allowing an adequate LEV effect and regulation of glutamate release. This diminishes neuronal hyperexcitability, causing a reduction of seizures.

In contrast, we did not observe changes in VGAT or SV2A-VGAT expression levels (except in the CA1 oriens). This suggests that the GABAergic system preserves normal operation in an early phase of the pilocarpine model of TLE. This is relevant since several reports have shown preferential SV2A function on GABAergic neurotransmission [33], even during epilepsy [12,30,32]. Moreover, LEV significantly increased GABA levels after high-K^+^ depolarization in the dorsal hippocampus using a model of TLE [34]. Thus, SV2A may be particularly important for the GABAergic system in the hippocampal formation; but may not be associated with the response to LEV treatment under our conditions.

## 5. Conclusions

The response to LEV treatment in an early stage of the pilocarpine model of TLE is related to differential SV2A distribution in excitatory versus inhibitory synapses, not its total expression. Particularly, SV2A expression in glutamatergic terminals is likely a key element for the response to LEV. On this basis, changes in SV2A expression should be considered according to the type of neurotransmission affected. The next step is to evaluate SV2A expression and distribution in the epileptic hippocampus when the disease is more advanced. Other synaptic vesicular proteins should be included, since their interaction could be involved in the response to LEV treatment.

## Figures and Tables

**Figure 1 brainsci-11-00531-f001:**
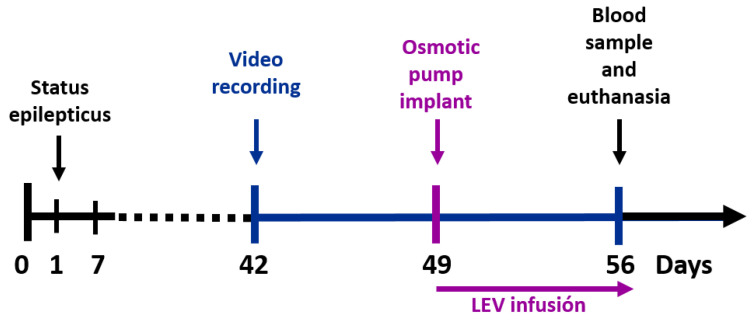
Experimental design. Status epilepticus (SE) was induced on day 1. Subsequently, spontaneous recurrent seizures were recorded before (days 42–48) and during (days 49–56) LEV treatment. A blood sample and the brain were obtained after humane euthanasia. LEV, levetiracetam.

**Figure 2 brainsci-11-00531-f002:**
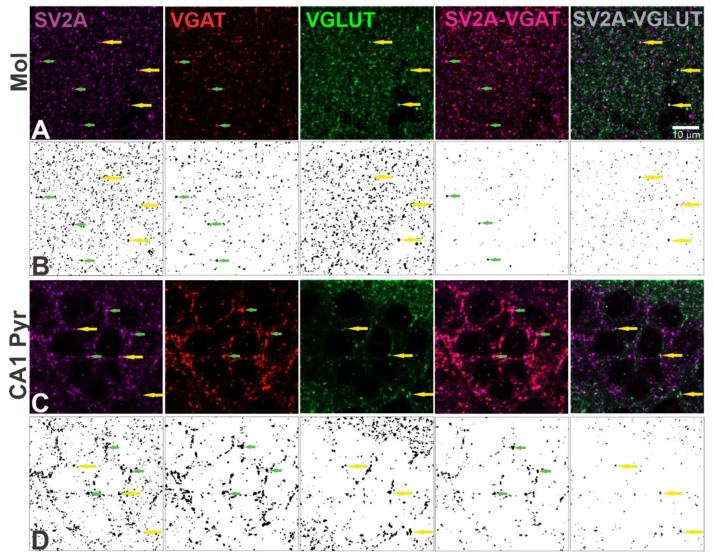
Example of immunofluorescent quantified puncta, using the Fiji plugin Synapse Counter, in high magnification images. (**A**,**C**) Confocal images of the molecular (Mol) and CA1 pyramidal (CA1 Pyr) layers with SV2A (purple), VGAT (red) or VGLUT (green) puncta, as well as SV2A-VGAT (magenta) or SV2A-VGLUT (gray) co-localized puncta. (**B**,**D**). Black and white images of puncta detected with Synapse Counter; each point corresponds to an immunofluorescent punctum. In each image, green arrows show SV2A-VGAT co-localized puncta, yellow arrows show SV2A-VGLUT co-localized puncta (scale bar = 10 µm).

**Figure 3 brainsci-11-00531-f003:**
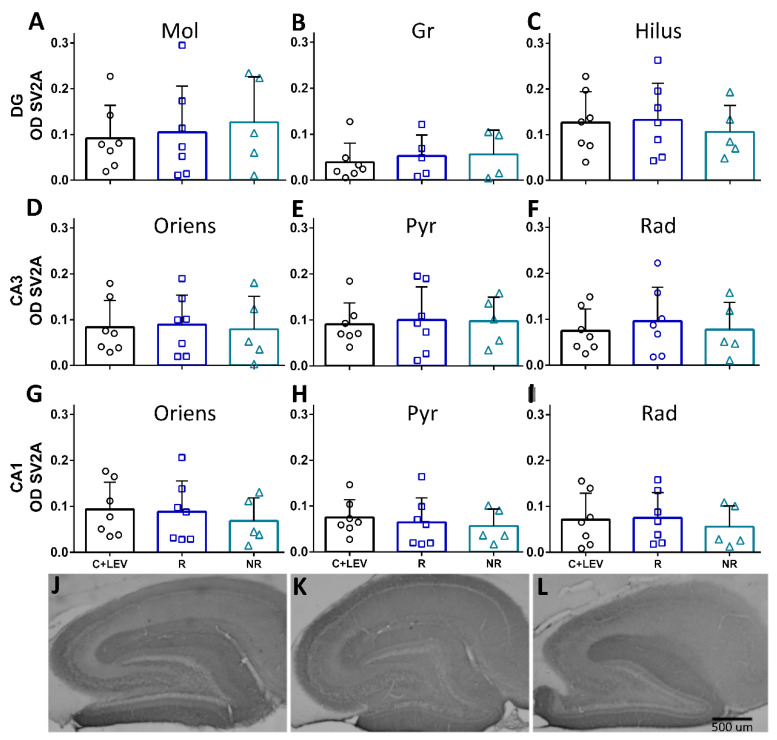
SV2A optical density (OD) after calibration and background subtraction. (**A**–**C**) Graphs of SV2A OD in the dentate gyrus (DG). (**D**–**F**) Graphs of SV2A OD in the CA3. (**G**–**I**) Graphs of SV2A OD in the CA1. The data are shown as mean ± SD (**J**–**L**). Representative microphotographs of SV2A OD in C+LEV (**J**), R (**K**), and NR (**L**) rats (scale bar = 500 µm).

**Figure 4 brainsci-11-00531-f004:**
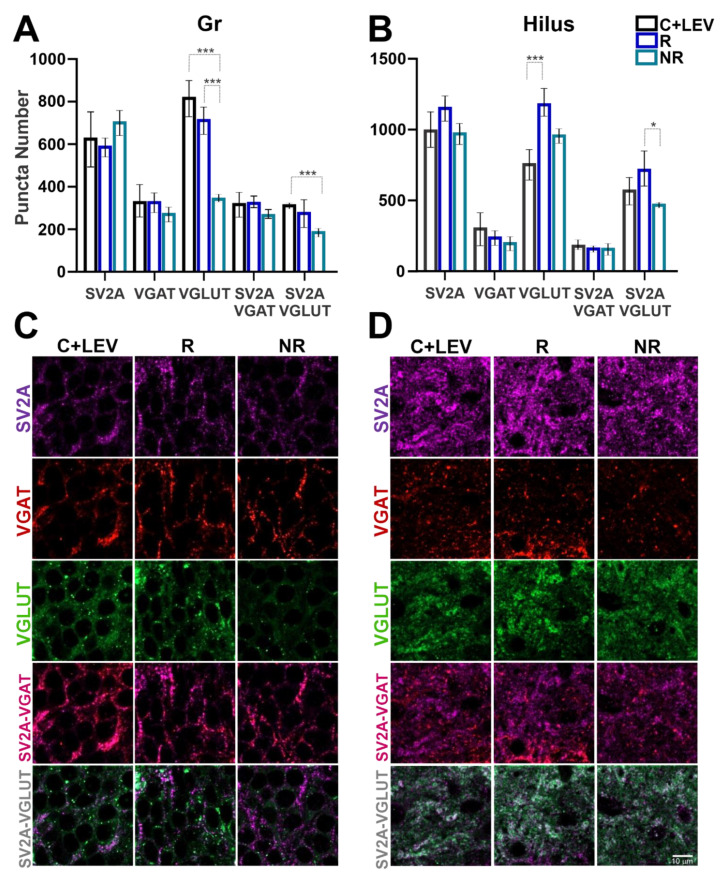
Quantification of SV2A, VGAT, VGLUT, SV2A-VGAT or SV2A-VGLUT puncta in the dentate gyrus granular layer (Gr) and hilus. (**A**,**B**) Graphs showing protein puncta (mean ± SD) in confocal images (50 × 50 µm) and their co-localization in C+LEV, R and NR animals. * *p* ≤ 0.05, *** *p* ≤ 0.001 (**C**,**D**) Confocal images of immunofluorescent SV2A (purple), VGAT (red) and VGLUT (green) puncta in the Gr and hilus, as well as SV2A-VGAT (magenta) or SV2A-VGLUT (gray) co-localized puncta in control plus levetiracetam (C+LEV), responder (R) and non-responder (NR) rats.

**Figure 5 brainsci-11-00531-f005:**
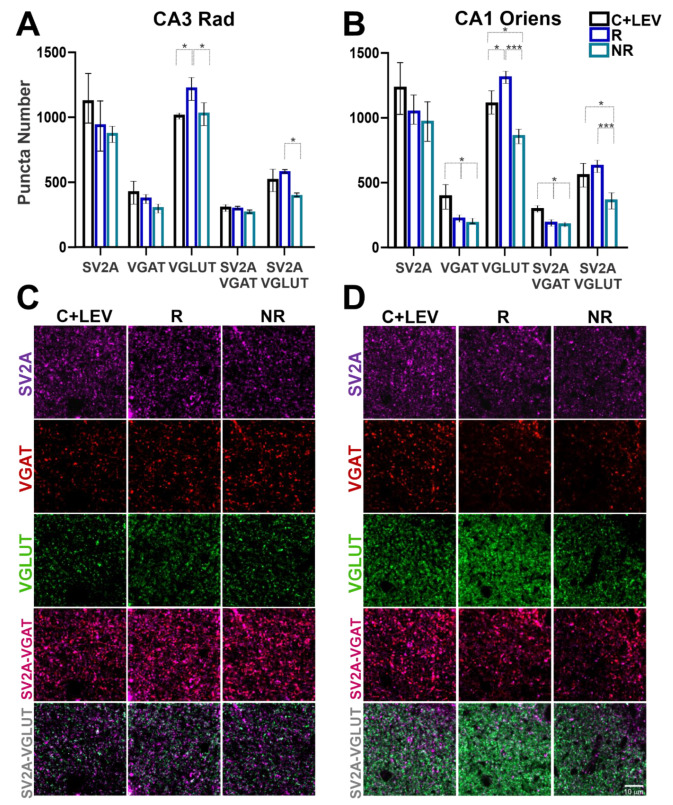
Quantification of SV2A, VGAT, VGLUT, SV2A-VGAT or SV2A-VGLUT puncta in the CA3 radiatum (CA3 Rad) and CA1 oriens. (**A**,**B**) Graphs showing protein puncta (mean ± SD) in confocal images (50 × 50 µm) and their co-localization in C+LEV, R and NR rats. * *p* ≤ 0.05, *** *p* ≤ 0.001 (**C**,**D**) Confocal images of immunofluorescent SV2A (purple), VGAT (red) and VGLUT (green) puncta in the Rad and oriens, as well as SV2A-VGAT (magenta) or SV2A-VGLUT (gray) co-localized puncta in control plus levetiracetam (C+LEV), responder (R) and non-responder (NR) rats.

**Table 1 brainsci-11-00531-t001:** Experimental conditions for each group.

Group	Sample Size (n)	SE Induced	Osmotic Pump Implanted
Control **C**	6	NO	NO
Control + Levetiracetam **C+LEV**	7	NO	YES
Epileptic **EPI**	6	YES	NO
Responders **R**	7	YES	YES
Non-responders **NR**	5	YES	YES

SE, Status epilepticus; C, control; LEV, levetiracetam; EPI, epileptic group; R, responders; NR, non-responders.

**Table 2 brainsci-11-00531-t002:** Effect of levetiracetam treatment on spontaneous recurrent seizures in epileptic rats.

Group	Pretreatment	During LEV Treatment
R	5.0 (3.0–7.0)	0.0 ^+^ (0.0–1.0)
NR	4.0 (2.5–28.5)	4.0 ** (2.5–16.5)

Number of SRS/week (medians, 25th–75th percentiles in parentheses) in levetiracetam treated epileptic rats. ** *p* ≤ 0.01 compared to R group during LEV treatment (Mann–Whitney rank-sum test), + *p* ≤ 0.05 compared to pretreatment in the same (R) group (Wilcoxon signed-rank test). SRS, spontaneous recurrent seizures; LEV, levetiracetam; R, responders; NR, non-responders.

**Table 3 brainsci-11-00531-t003:** Serum levetiracetam concentration.

Group	Serum Levetiracetam Day 56 (µg/mL)
C+LEV	59.74 ± 14.49 (36.5–75.6)
R	44.89 ± 14.06 (32–66.3)
NR	49.52 ± 18.56 (36.6–76.7)

Serum levels (mean ± SD, range in parenthesis) of levetiracetam in treated control (C+LEV) and epileptic animals (R, responders; NR, non-responders).

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
