# Peer review of "Synaptic Vesicle Protein 2A Expression in Glutamatergic Terminals Is Associated with the Response to Levetiracetam Treatment"

_brainsci, 2021, doi:10.3390/brainsci11050531_

Round 1

Reviewer 1 Report

The authors investigate SV2A expression in glutamatergic terminals is associated with the response to levetiracetam treatment.

All the sections of the paper are well written> the introduction offers a good background for the purpose of the study, the MM is detailed and the discussion section is sustained by the results.

Reviewer 2 Report

The work presented by the authors is very necessary and important in the scientific field, with impact in epilepsy disease knowledge for achievement of the treatment. However some improvements must be done in the presented work, improvements which are suggested below.

Abstract:
Please, the text of abstract although well organised needs to be polished.
"VGAT and VGLUT" - The first time the authors use an abbreviation, it should be written out completely.
"SV2A-VGLUT co-expression" - What is this about? Please, be as clear as possible.
"the responder group showed greater" - If the authors could use measures expressed with numbers and/percentages, it is always better than "greater".
Keywords: "... vgat; vglut..." - Please use always the same format to write the same names.

Introduction:
Improve this description, please: "However, some reports show that SV2A participates in the synaptic vesicle cycle (exocytosis and endocytosis; [3])."
"regulates the expression and trafficking of..." - Which is the mechanism involved?
"synaptic release" - Release of what?
"neurotransmitter release" - Which one is it talking about?
If the authors concluded in the abstract with the results that "SV2A expression was not modified during epilepsy.", the authors could say in the introduction that "SV2A is the target of the antiepileptic drugs levetiracetam (LEV) and brivaracetam [1, 12, 13], constituting their primary mechanism of action." (...) "clinical response to LEV has been associated to SV2A expression levels [25]. (...) LEV treatment shows overexpression of Sv2a and other genes encoding proteins involved in vesicle trafficking" - Is there something that is not according with your results? Discuss it please. Or construct the text in a different way, since the discussion should be made in the discussion section.

Methods:
"Wistar rats (Envigo, México)" - Is this the supplier? If it is, it should say purchased from..., or some thing like that.
"Institutional Animal Welfare Committee (INP-064-2015)." - The licence referred seems old (2015) in relation to these results presented. Is this still the licence for your work?
Animals: Please include the age and body weight of the animals used.
How many animals were used in each group?
The toxicity of LEV is already tested in literature?
Is reference 34 the study used for performing the metodology? "no reduction in their seizures although they re-ceived LEV [34]." (...) "(> 50 %), they were classified as responders; otherwise (< 50 % seizure reduction) were classified as non-responders."- Perhaps the authors should consider another model or adjust the model described in reference 34.

Improve Table 1. "Control + Levetiracetam C+LEV" should be in the same line?

Materials used should be listed with the codes, purity, suppliers, etc., and not like "bromide (1 mg/kg i.p.; S8502); (30 mg/kg i.p.; P6503); (5 mg/kg i.m.; PISA); LEV (200 mg/kg i.p.; UCB Laboratories); sucrose (S7903); phosphate buffer (PB 0.1 M, pH 7.4, S8282 and S7907); etc"
"PlayBack program" ; GraphPad Prism (v. 9.0.0)"- Please provide the supplier and the complete reference of the program.
"Osmotic pumps (Alzet 2ML1)" - Please, complete the information about pumps, supplier, model, city, etc.
"Mikro 12-24, Hettich" - Please, provide the country/city.
"filtered (Corning® 0.45 μm)"; "(Sofloran Vet, PISA), cryostat (Cryocut 1800)" - Please improve this material reference.
"card (Whatman 903; [33])." ; "paraformaldehyde (158127)." - What does this mean exactly?
Figure 2 is in fact a result, it should be in the results section.
"LEV concentration in blood were expressed as mean ± S.E.M." - The work does not have enough data to use SEM, in this case SD should be used. The same for Figure 3,5.
This section need much attention.

Results:
"Mol and CA3 Rad" - The first time abbreviations are used, they should be explained.
"In this study, VGLUT increased... oriens of NR rats" - What is it (oriens)?

Reviewer 3 Report

The present manuscript entitled "SV2A expression in glutamatergic terminals is associated with the response to levetiracetam treatment" is interesting and informative. 

The authors have the great merit of having treated a complex topic in a clear, simple and explanatory way. Each section of the article is excellently balanced and clear. The methods are clearly described and the results are interpreted correctly. The references are adequate and targeted on the subject.
I believe that this article is an excellent tool to be used also for educational purposes for the mechanisms of epileptogenesis.

Only some minor spelling mistakes and some lack of definitions of abbreviations (eg. VGAT).

Round 2

Reviewer 2 Report

Figure 4 and 5 - A and B needs to be improved for better interpretation. 

References section - Reference 54 has a different format from all of other references.
